# Daily Soil Moisture Retrieval by Fusing CYGNSS and Multi-Source Auxiliary Data Using Machine Learning Methods

**DOI:** 10.3390/s23229066

**Published:** 2023-11-09

**Authors:** Ting Yang, Jundong Wang, Zhigang Sun, Sen Li

**Affiliations:** 1CAS Engineering Laboratory for Yellow River Delta Modern Agriculture, Institute of Geographic Sciences and Natural Resources Research, Chinese Academy of Sciences, Beijing 100101, China; sun.zhigang@igsnrr.ac.cn; 2Shandong Dongying Institute of Geographic Sciences, Dongying 257000, China; 3Key Laboratory of Ecosystem Network Observation and Modeling, Institute of Geographic Sciences and Natural Resources Research, Chinese Academy of Sciences, Beijing 100101, China; wangjd.20b@igsnrr.ac.cn; 4College of Resources and Environment, University of Chinese Academy of Sciences, Beijing 100049, China; 5National Meteorological Center, China Meteorological Administration, Beijing 100081, China

**Keywords:** CYGNSS, soil moisture, data fusion, land cover, GBRT

## Abstract

The Cyclone Global Navigation Satellite System (CYGNSS), a publicly accessible spaceborne Global Navigation Satellite System Reflectometry (GNSS-R) data, provides a new alternative opportunity for large-scale soil moisture (SM) retrieval, but with interference from complex environmental conditions (i.e., vegetation cover and ground roughness). This study aims to develop a high-accuracy model for CYGNSS SM retrieval. The normalized surface reflectivity calculated by CYGNSS is fused with variables that are highly related to the SM obtained from optical/microwave remote sensing to solve the problem of the influence of complicated environmental conditions. The Gradient Boost Regression Tree (GBRT) model aided by land-type data is then used to construct a multi-variables SM retrieval model with six different land types of multiple models. The methodology is tested in southeastern China, and the results correlate very well with the existing satellite remote sensing products and in situ SM data (R = 0.765, ubRMSE = 0.054 m^3^m^−3^ vs. SMAP; R = 0.653, ubRMSE = 0.057 m^3^ m^−3^ vs. ERA5 SM; R = 0.691, ubRMSE = 0.057 m^3^m^−3^ vs. in situ SM). This study makes contributions from two aspects: (1) improves the accuracy of the CYGNSS retrieval of SM based on fusion with other auxiliary data; (2) constructs the SM retrieval model with multi-layer multiple models, which is suitable for different land properties.

## 1. Introduction

Studying soil moisture (SM) with high accuracy is crucial for land–air interactions, affecting the climate system by influencing processes such as evapotranspiration and water transport at the land surface [1,2]. Accordingly, this highlights the necessity of retrieving SM data with high accuracy. SM can be retrieved by optical remote sensing missions, such as Moderate Resolution Imaging Spectroradiometer (MODIS), microwave remote sensing missions, such as Soil Moisture Active Passive (SMAP), Sentinel-1 (Sentinel-1) [3,4,5]. It is noted that optical remote sensing is seriously affected by cloud cover; passive microwave remote sensing has a low spatial resolution (9~40 km), and active microwave remote sensing has a low temporal resolution (≥6 days).

With the rapid development of the spaceborne Global Navigation Satellite System Reflectometry (GNSS-R) remote sensing technology, this technique provides a new, useful, and supplemental tool for SM retrieval at a regional scale. Spaceborne GNSS-R is an emerging remote sensing technique that exploits the capability of GNSS satellites in a bistatic radar configuration, and collects the Earth’s reflected signals by specially designed receivers. This technology has the unique advantages of a short revisit period, the ability to provide a cost-effective alternative, and a high spatial–temporal resolution. The CYGNSS mission consists of a constellation of eight satellites orbiting in the same plane at an altitude of approximately 510 km and an orbit inclination of 35°. Each micro-satellite is equipped with a specially designed GNSS receiver named SGR-ReSI. The GNSS-R payload on the CYGNSS mission can receive signals of the GPS L1C/A band, while the BeiDou, Galileo, and GLONASS data are not yet available for use. CYGNSS can obtain the surface retrievals at the level of a few hours, i.e., 2.8~7 h per day; in terms of spatial resolution, the minimum spatial footprint is ~3.5 km × 0.5 km for coherent scattering from smooth surfaces [6,7]. Moreover, the CYGNSS uses the L-band (1–2 GHz), which is relatively sensitive to surface SM retrieval. 

The spaceborne GNSS-R SM retrieval methods used at present can mainly be categorized into two types, i.e., empirical and data-driven methods. The empirical method takes SMAP SM or in situ SM as the reference data, establishing the conversion relationship between spaceborne GNSS-R surface reflectivity (SR) and the reference SM [7,8,9]. At present, SMAP is the most widely used regional-scale SM data product for comparison with CYGNSS data, and it operates in the same band (i.e., L-band) as CYGNSS, making a direct comparison of the SM data meaningful [7,8]. Due to the high heterogeneity of SM, SM can reflect the surface conditions under the effect of climatic, meteorological, and hydrological factors, e.g., vegetation water content (VWC), land surface temperature (LST), slope, surface roughness, and precipitation. Subsequently, the data-driven method is derived based on the empirical method, which mainly combines spaceborne GNSS-R with multi-source remote sensing auxiliary data using machine learning (ML) models to improve the accuracy of SM retrieval [10,11,12,13,14,15,16,17]. Yang et al. (2020) fused the CYGNSS, TDS-1 data, and multi-source remote sensing data based on the BP neural network to retrieve SM in southeastern China at 36 km resolution [10]. Senyurek et al. (2020) comparatively analyzed the validation of the advantages in CYGNSS SM retrieval based on a variety of common ML algorithms (ANN, RF, and SVR) [11]. To make the CYGNSS SM applicable to different land types with high accuracy Jia et al. (2021) estimated the global SM with a spatial of 36 km using ML regression aided by a pre-classification strategy [12]. 

This study aims to develop a high-accuracy model for CYGNSS SM retrieval, based on the approach described by [13]. First, the normalized CYGNSS SR is retrieved using the coherent model. Then, normalized CYGNSS SR is fused with the variables highly related to the SM obtained from optical/microwave remote sensing data, aiming to solve the problem of the influence of complicated environmental conditions. Finally, these parameters are combined through the Gradient Boosted Regression Trees (GBRT) model; here, multiple models with six different land cover types (i.e., cropland, forest, grassland, shrubland, wetland, and bare land) are built, and the results derived from each model are linked together. The result is validated with SMAP SM, ERA5 SM, and in situ SM.

## 2. Data and Methods

### 2.1. Data

#### 2.1.1. CYGNSS

CYGNSS consists of eight microsatellites with a global coverage of approximately from 38° N to 38° S. Each satellite carries a four-channel GNSS-R payload, for GNSS-R low-orbit measurements. The average revisit period is about 7.2 h [8]. Many researchers utilize the observations of Delayed Doppler Map (DDM) data from CYGNSS for SM retrieval. The CYGNSS receivers firstly calculate the satellite position and receiver position, based on ephemeris for the collected direct GNSS satellite signals and the reflected GNSS satellite signals, respectively; the specular reflection point position and Doppler delay of the reflected signals are then calculated to generate the DDM [6]. Level 1a data of CYGNSS V 3.1 version were used in this study. During the data preprocessing, the water bodies were masked, and only the land data were retained. The CYGNSS data in 2020 and 2021 were chosen as the training data and testing data, respectively. In addition, to synchronize with the in situ SM, we simultaneously tested the CYGNSS in July 2018. Since the CYGNSS V 3.1 version was unavailable until August 2018, the V 2.1 version of CYGNSS data was compared with in situ SM. The CYGNSS data were finally resampled to a 9 km Equal-Area Scalable Earth (EASE) Grid 2.0 cell. 

#### 2.1.2. SMAP

The SMAP operates at L-band, with an incidence angle of about 40°. Due to the satellite orbit design, there are gaps in the daily SMAP SM product, and the revisit time ranges from 2 to 3 days [18]. As mentioned before, SMAP SM is commonly used for comparison with CYGNSS SR data in empirical or data-driven methods to convert the CYGNSS SR to the final SM. 

The SMAP SM data were obtained from the National Snow and Ice Data Center (NSIDC) at the L3 level with a spatial resolution of 9 km. In this study, the “retrieval_qual_flag” variable in the data product was utilized for SM quality control. At the same time, the daily ascending and descending data were averaged as the final daily SM. SMAP values larger than 0.6 m^3^m^−3^, and less than 0.04 m^3^m^−3^ are excluded. To synchronize with the CYGNSS data, the SMAP data of 2020 were selected as the model target data, and the SMAP data of July 2018 and 2021 were used as the testing data, respectively.

#### 2.1.3. ERA5-Land

The CYGNSS and SMAP satellites both work at the L-band. To further verify the accuracy of the retrieved SM in this study, ERA5-Land SM data were selected as an independent data source for a cross-comparison of the results. ERA5-Land data were obtained from the European Center for Medium-Range Weather Forecasts (ECMWF). This is a reanalyzed dataset consisting of a combination of the modeled data and in situ observations. The spatial resolution of ERA5-Land is 0.1°, and the temporal resolution is 1 h [19]. This study averaged the hourly ERA5-Land SM to obtain its daily average SM. Second, to match with the CYGNSS SM, this study resampled the ERA5-Land SM to 9 km based on the nearest neighbor interpolation method.

#### 2.1.4. In-Situ SM Data

The in situ SM data from 596 stations collected on 1 July, 11 July, and 21 July 2018, were provided by the China Meteorological Administration (Figure 1). These data are currently not publicly available. Data were collected to a depth of 10 cm and then averaged at timepoints ranging from hourly to daily. It is worth noting that the L-band signals are relatively sensitive to the surface SM (~5 cm depth); thus, there is a different sensitivity depth between in situ SM and CYGNSS SM. Considering the complex geographical conditions in China, the selected stations were distributed in six provinces of China (i.e., Henan, Hunan, Shandong, Jiangxi, Sichuan, and Yunnan), with different land types and topographic distributions.

#### 2.1.5. Auxiliary Data

Significant spatial differences in topography and vegetation changes were also considered in this study to select auxiliary variables affecting SM. Variables such as climate, elevation, slope, precipitation, soil, and vegetation characteristics can also influence SM. The vegetation index can provide information on the growth status of vegetation; LST is a good indicator to characterize the energy balance of the land surface, and can indirectly reflect the SM status [20,21]. In addition, precipitation strongly influences soil infiltration, runoff, and evapotranspiration, which are the main control factors affecting SM [22]. Topographic factors, such as slope, aspect, and elevation, influence SM changes and distribution patterns [23]. The surface albedo was estimated using MCD43A3 reflectance by the method proposed by [24]. Table 1 summarizes the auxiliary data information used in this study. Here, slope and aspect are calculated based on the NASA Shuttle Radar Topographic Mission (SRTM) 90 m Digital Elevation Models (DEM) data [23]. It is worth mentioning that the VWC derived from SMAP is retrieved based on the empirical method using the Normalized Difference Vegetation Index (NDVI) of the optical data, and thus there is no strong correlation between the data and SMAP SM. It has been proved in [13] that the surface roughness variable has less influence on the SMAP SM retrieval than the vegetation, so the impact of VWC and surface roughness on the SMAP data can be ignored. 

### 2.2. Methodology

The overall flowchart for this study is shown in Figure 2. The main steps are summarized as follows: (1) the normalized CYGNSS SR is obtained using the coherent model; (2) quality control of auxiliary data: variables highly related to the SM obtained from optical/microwave remote sensing data are matched up, aiming to solve the problem of the influence of complicated environmental conditions; (3) SM retrieval: this step is executed to build the data-driven model for retrieving SM from input parameters through the GBRT model, where the GBRT model contains six models aided by different land types (i.e., cropland, forest, grassland, shrubland, wetland, and bare land); (4) validation: the result is evaluated by comparison with SMAP SM, ERA5-Land SM, and in situ data.

Three main differences exist between the algorithm presented by Yang et al. (2020) [13] and the one presented here. The first is that the proposed method uses normalized CYGNSS SR as the proxy of CYGNSS SM to eliminate the effect of incidence angle. Second, this retrieval algorithm uses six sub-models with different land cover types. Third, the SM resolution obtained by the two algorithms is different, i.e., 9 km vs. 36 km. 

#### 2.2.1. Normalized CYGNSS SR Retrieval

The CYGNSS approaches for retrieving SM should account for the dominance of coherent or incoherent effects. The coherent method assumes that the coherent component dominates the land reflections, but dense vegetation and a rough terrain can cause incoherent scattering [4,6,7,10,14]. It should be pointed out that the dense-vegetation- and rough-terrain-induced incoherence scattering mentioned here do not conflict with our study correcting the vegetation effects. The discussion of vegetation-induced incoherence aims to determine how SR is derived, since the methods of deriving the SR from coherent and incoherent scattering signals are totally different. Through the GlobeLand30-2020 land classification data statistics [25], the proportion of CYGNSS located in densely vegetated areas accounts for only forested areas accounted for ~7.53%; therefore, this study assumes that the reflected signals are dominated by the coherent scatterings. 

For the CYGNSS data preprocessing step, firstly, the data with a signal-to-noise ratio of less than 0 are filtered; secondly, data with incidence angles greater than 65° are removed; thirdly, the variable of “quality control” (e.g., spacecraft attitude error, blackbody DDM direct signal, and GPS equivalent isotope radiated power) is used to select the good data acquisition. The SR (dB) can be calculated as [7,14]:(1)$$\begin{array}{cc} \Gamma_{r,eff}(dB)= & {10\log P}_{r}+20\log(R_{ts}+R_{sr})-10\log G_{t}-20\log\lambda \\ & -10\log P_{t}-10\log G_{r}-20\log\left(cos\theta \right) \end{array}$$
where *Γ_r,eff_* is the SR; *P_r_* represents the peak DDM power minus the noise; *R_ts_* represents the distance between the transmitter and the specular reflection point; *R_sr_* represents the distance between the specular reflection point and the receiver; λ is the GPS wavelength; *Gt* is the gain of the GPS transmitting antenna; *Gr* is the gain of the receiving antenna; *P_t_* is the transmitted power. The above observables are available from the CYGNSS L1a V 3.1 and V 2.1 data. 

In addition to the effect of the soil dielectric constant, SR is also affected by the incidence angle, as in Equation (1). To eliminate the effect of the incidence angle, and retain only the effect of the soil dielectric constant, the incidence angle of SR is normalized to 0° here. Previous studies have investigated the reflected signals of CYGNSS dependence on the incidence angle [9,26]. Thus, to eliminate the effect of incidence angle in CYGNSS SR, the CYGNSS SR is calibrated as follows:$$\Gamma_{r,eff}\left(0,dB\right)=\Gamma_{r,eff}\left(\theta ,dB\right)/f\left(\theta \right)$$
where $\Gamma_{r,eff}\left(0,dB\right)$ is the normalized SR, hereafter named SR_n_*;*
f(θ) is the modified function. Here, f(θ) is not a simple expression; it is related to the incidence angle. The f(θ) value shows a nonlinear decreasing trend as the incidence angle increases. When the incidence angle is 0°, the value of f(θ) is 1; when the incidence angle is 65°, the f(θ) has a value of about 0.8. We previously simulated the value of f(θ) varying with the incidence angle, and the value of $f\left(\theta \right)$ can be referred to in [26]. Subsequently, the CYGNSS SR is gridded to 9 km spatial resolution.

#### 2.2.2. SM Retrieval Model Construction

This study comprehensively considers the effects of eight environmental and land auxiliary variables (i.e., VWC, precipitation, LST, surface albedo, elevation, surface roughness, aspect, and slope) on SM. CYGNSS SR_n_ (spatial resolution of 9 km) is used as the reference data, and other data are all resampled to 9 km using the nearest-neighbor interpolation method. Jia et al. (2021) proposed a pre-categorization strategy to minimize the influence of different land types [12]. This study thus constructs multiple models considering the complex geographical environment. The GBRT model is used to construct the model aided by the land cover data separately for collaborative computation (Figure 2). The GBRT is a typical ML method; it combines basic regression trees and boosting techniques to perform nonlinear regression and classification [26]. Theoretically, the GBRT uses the classification and regression trees as weak classifiers, and uses the boosting techniques for iterative training [27]. In this study, the “sklearn” library in Python is used.

Based on the resampled 9 km GlobeLand30 land-type data [27], the input data are divided into six classes according to land type (cultivated, forest, grass, shrub, wetland, and barren). The whole model is then divided into six models for training to reduce the effect of land properties and improve the model’s adaptability to complex surfaces.

The steps of the model are as follows:(1)The area is divided into six land types according to the GlobeLand30 data. All auxiliary data are resampled to 9 km. The feature data of the same land types are then extracted to construct multiple models.(2)Multiple models of six different land types are built to extract SM features. Then, the feature fusion layer is constructed to connect all nodes. The optimal structure of the model is determined through repeated testing.(3)The dataset is divided into a training set and a testing set to verify the accuracy of the model. The training set is used to adjust the variables of the GBRT model, and the testing set is used to test the performance. The fusion model process is shown in Figure 3.

#### 2.2.3. Error Metrics

Two statistical error metrics, i.e., the correlation coefficient (R), and the unbiased root mean square error (ubRMSE), are used to provide quantitative analysis between the CYGNSS SM and SMAP SM. The formulas for these metrics are shown:(2)$$R=\frac{\sum \left(E_{i}-\bar{E}\right)\left(O_{i}-\bar{O}\right)}{\sqrt{{\sum \left(E_{i}-\bar{E}\right)}^{2}{\left(O_{i}-\bar{O}\right)}^{2}}}$$
(3)$$ubRMSE=\sqrt{\frac{1}{n}\sum_{i=1}^{n}{\left[\left(E_{i}-\bar{E}\right)-\left(O_{i}-\bar{O}\right)\right]}^{2}}$$
where n is the number of observations; $O_{i}\;$is the reference SM; E_i_ is the CYGNSS SM.

## 3. Results

### 3.1. Quality Control of Auxiliary Variables

Determination of the input features is an important step in the proposed model. The importance of the input variables to the SM is calculated using the function of “permutation_importance” from the sklearn module of Python. Permutation importance is a method used to assess the importance of features based on model prediction results. It obtains the value of feature importance by randomly disrupting individual features to obtain the extent of each feature variation’s effect on model accuracy [27]. The value is then measured to determine the variable of importance to the target data, i.e., SMAP SM. The importance values of the above eight variables (i.e., CYGNSS SR, VWC, slope, surface roughness, precipitation, surface albedo, land surface temperature, aspect, and elevation) are listed in Table 2. In addition, to assess the contribution of CYGNSS in SM retrieval, the CYGNSS SR is also included in calculating importance values, along with auxiliary variables. The testing data were collected from 1 July 2021 to 31 July 2021 in southeastern China, and the target data were SMAP SM. As illustrated in Table 2, the CYGNSS SR shows the highest value of feature importance with SMAP SM. The feature importance of VWC, precipitation, and LST is greater than 0.1. Conversely, the values of elevation and aspect are less than 0.02. Thus, it can be concluded that the elevation and aspect are less correlated with the SM. In this study, except for the CYGNSS SR, six auxiliary variables with feature importance values greater than 0.02 were selected to construct the SM retrieval model, i.e., VWC, slope, surface roughness, precipitation, surface albedo, and LST. It should be noted that the test data presented here are relatively short compared to the long-term data input in the GBRT model, which may lead to fluctuations in the relative weights of the feature importance values. 

### 3.2. Comparison with SMAP SM

To validate the retrieval accuracy of CYGNSS SM, the spatial distributions of the CYGNSS SM, and SMAP SM on 18 January 2021, are shown in Figure 4a,b. The SMAP is three days averaged to a final value, aiming to fill the gap in the data. The result of CYGNSS SM minus SMAP SM is shown in Figure 4c. The CYGNSS SM shows a good consistency of fluctuations with SMAP SM, and the CYGNSS SM is close to the SMAP SM result. The result shows that the SM values are larger in the central region, with a variation range between 0.3 and 0.5 m^3^ m^−3^, and smaller values are seen in the north, with a variation range between 0 and 0.2 m^3^ m^−3^. The delta value varies in the range −0.19~0.15 m^3^ m^−3^, with a mean value of −0.003 m^3^ m^−3^, and the delta value varies in the central region. 

As shown in Figure 5, the R and ubRMSE are introduced to compare the CYGNSS SM with SMAP SM in 2021. Generally, the spatial patterns of both indices are good, with a mean R = 0.765 m^3^m^−3^, and ubRMSE = 0.054 m^3^m^−3^. The performance of R is relatively good, with more than 60% of the R-values being larger than 0.6. In terms of ubRMSE, most of the region has ubRMSE values less than 0.05 m^3^m^−3^, while the northeast and west regions are greater than 0.03 m^3^ m^−3^. The incoherent scattering caused by dense vegetation and high surface roughness may lead to this phenomenon.

### 3.3. Comparison of with ERA5-Land SM

Figure 6 shows the spatial distribution of R and ubRMSE of CYGNSS SM and ERA5-Land SM in 2021 based on a daily scale. In most study areas, CYGNSS SM correlates well with ERA5-Land SM, with an average value of R = 0.653 and ubRMSE = 0.057 m^3^ m^−3^ due to the different spatial mismatches in the measurement depth between ERA5-Land SM detection (an effective depth of 0–7 cm) and CYGNSS L-band frequency signal penetration (an effective depth of 0–5 cm). In addition, ERA5 SM is reanalysis model-derived data and may not accurately capture SM dynamics. Figure 6 shows a lower correlation in the overlap area than SMAP SM. 

### 3.4. Comparison with In Situ SM

This study utilizes the in situ SM data from July 2018 in southeastern China to further validate CYGNSS SM’s retrieval accuracy. The CYGNSS SM, located less than 0.45 km from this site, was selected during the observation period. Here, 5766 in situ SM were used and 786 in situ SM were used. The results of the density plots are shown in Figure 7. Specifically, the data agree well with an R greater than 0.65, and RMSE = 0.057 m^3^m^−3^. This indicates that the CYGNSS SM provides fine-scale information on SM and can capture variations in the observed SM. 

### 3.5. Comparisons with the GBRT Model without Land Cover Classification

To show the benefits of utilizing multiple models to include land cover information, Figure 8 shows the average SM validation results for six land cover types from January to June, 2021. The R values are calculated from CYGNSS SM and SMAP SM, with an average value of 0.675. As shown in Figure 8a, the R values are lower in forested areas, and this phenomenon may be due to the larger proportion of incoherent reflective signals in these areas. The R values of grassland and barren areas are higher, with values greater than 0.65. Figure 8b shows the results of the ubRMSE. The ubRMSE of all land types is less than 0.07 m^3^ m^−3^, and the average value is 0.053 m^3^ m^−3^. The results indicate that the retrieval model proposed in this study is relatively reliable.

To further prove the effectiveness of the model proposed by this study, this study compares the accuracy of the proposed model (GBRT_GL30_SM_) with the GBRT model without land cover classification (GBRT_SM_). As shown in Figure 8, the R-value of the GBRT__GL30_SM_ model is higher than that of the GBRT_SM_ model, and the mean value of R increases from 0.617 to 0.675 (~9.40% gain in R); in terms of the ubRMSE, the mean value of the ubRMSE decreases from 0.058 m^3^ m^−3^ to 0.053 m^3^ m^−3^ (~8.62% decrease in ubRMSE).

## 4. Discussion

The research community regarding the use of spaceborne GNSS-R to sense SM has been facing difficulties in removing the effects of complex terrain. This methodology differs from other studies in that it retrieves SM from using multiple different land cover models constructed by the GBRT model, and shows an improvement in the accuracy of SM. In addition, the proposed methodology provides a complementary method for estimating SM.

According to the presented results, the proposed framework has several limitations. First, the reliable SM datasets rely on the use of SMAP as a reference rather than retrieving SM from the mechanistic perspective; errors in SMAP SM products will lead to uncertainty in the final results. Moreover, since most signals are probably a combination of incoherent and coherent scattering, we assume the reflected signals are coherent, which may lead to errors in final SM retrievals. Therefore, the possible directions for progressively enhancing this methodology should focus on comprehensively considering the incoherent and coherent signal scattering.

## 5. Conclusions

This study combines CYGNSS data with other remote sensing data to address the effect of complicated environmental conditions. Using the GBRT model, aided by land-type data, a multi-variables SM retrieval model with six different land cover submodels was built. The results of this study are expected to support the SM retrieval method for in-orbit GNSS satellites.

The main conclusions of the study are summarized as follows:
(1)The method proposed in this study can retrieve SM with high accuracy, with R= 0.765 m^3^m^−3^, and ubRMSE = 0.054 m^3^m^−3^ compared to SMAP SM, and R = 0.653 and ubRMSE = 0.057 m^3^ m^−3^ compared to ERA5-Land SM, and with R = 0.691, and RMSE = 0.057 m^3^m^−3^ compared to the in situ SM.(2)The accuracy of the proposed model is improved compared to the model without land cover classification, with the R-value improved by ~9.40% and the ubRMSE value decreased by ~8.62%.

## Figures and Tables

**Figure 1 sensors-23-09066-f001:**
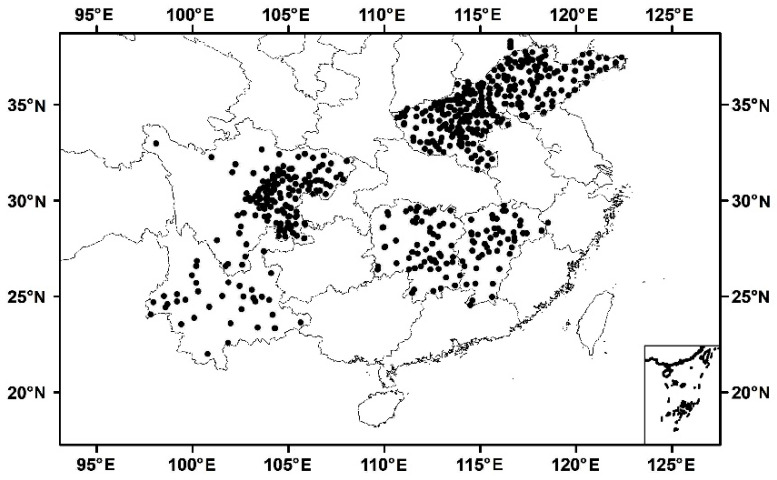
Distribution of in situ SM (1 July 2018).

**Figure 2 sensors-23-09066-f002:**
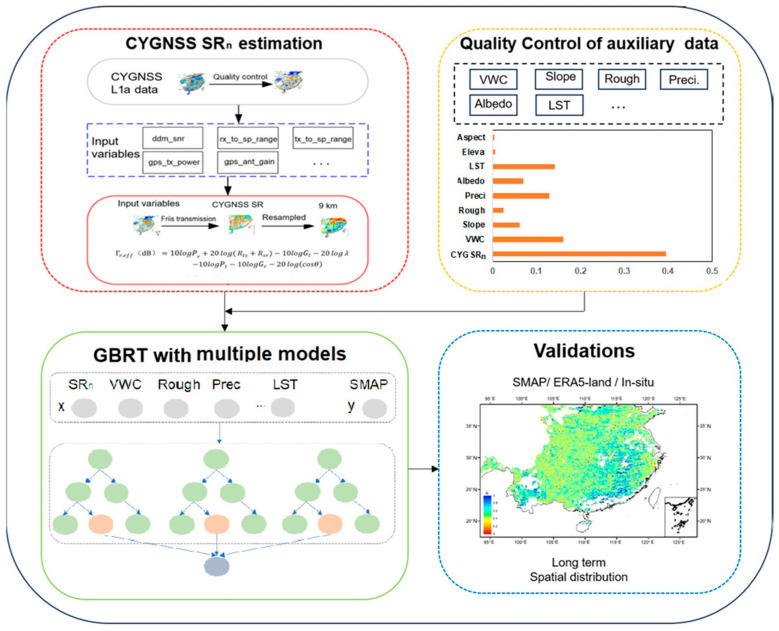
Flowchart of this study.

**Figure 3 sensors-23-09066-f003:**
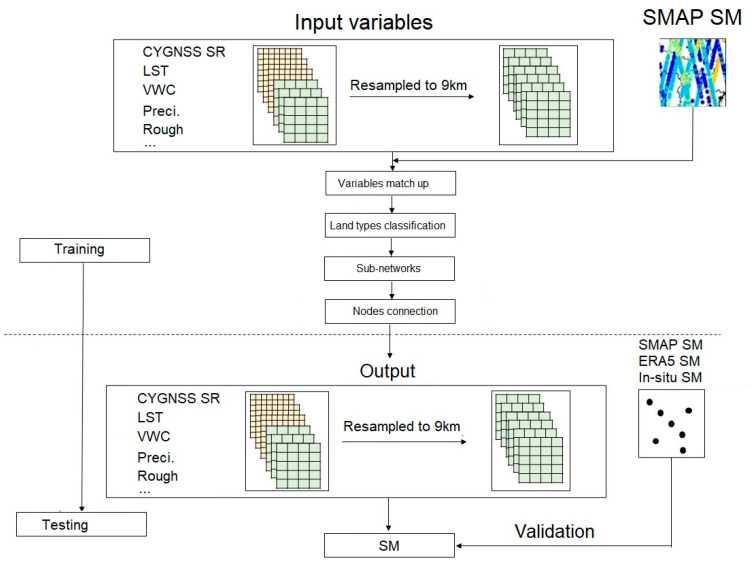
Flow chart of the fusion model.

**Figure 4 sensors-23-09066-f004:**
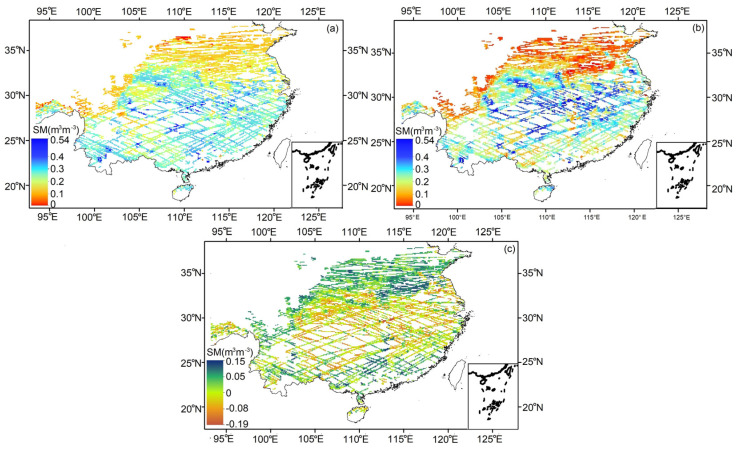
SM in southeastern China on 18 January 2021 (9 km). (**a**) CYGNSS SM, (**b**) 3-day averaged SMAP SM, (**c**) delta between the two data.

**Figure 5 sensors-23-09066-f005:**
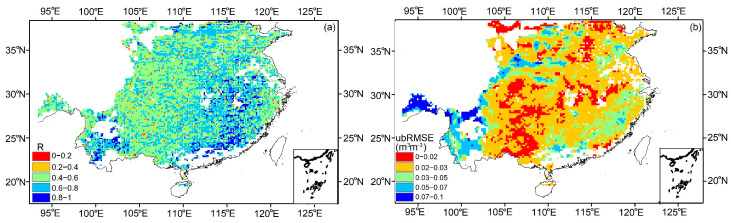
Comparison with SMAP SM for 2021 in southeastern China (9 km). (**a**) R, (**b**) ubRMSE.

**Figure 6 sensors-23-09066-f006:**
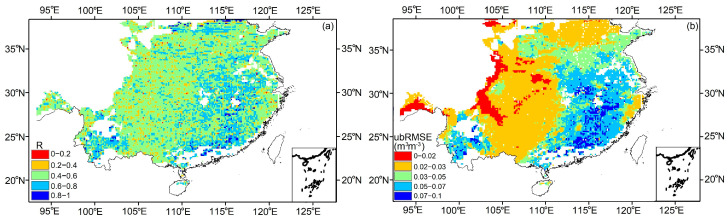
Comparison with the ERA5-Land SM in 2021 in southeastern China (9 km). (**a**) R, (**b**) ubRMSE.

**Figure 7 sensors-23-09066-f007:**
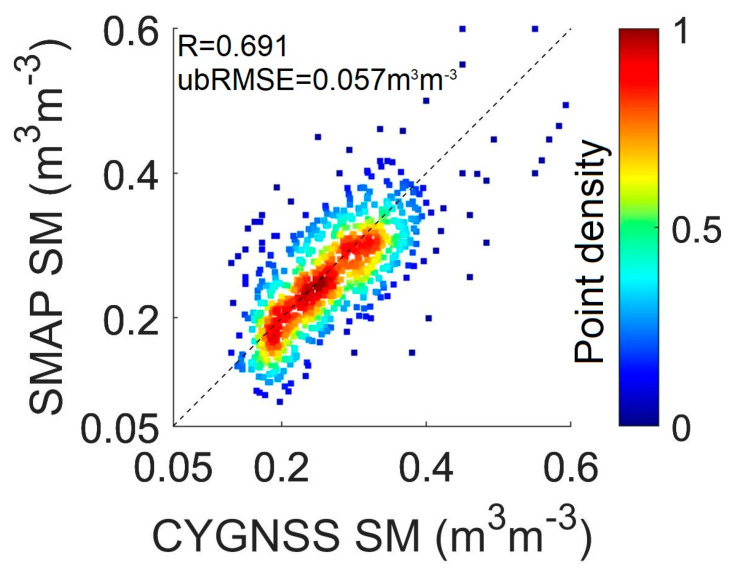
Comparison of CYGNSS SM with in situ SM in July 2018.

**Figure 8 sensors-23-09066-f008:**
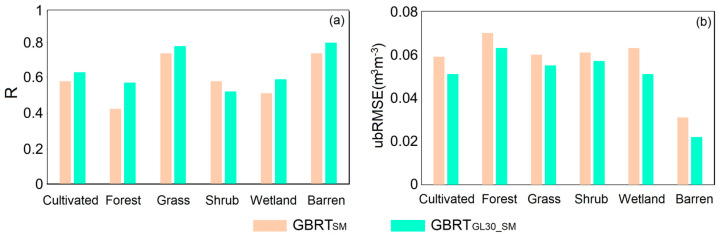
Accuracy of SM under different land types. (**a**) R, (**b**) ubRMSE.

**Table 1 sensors-23-09066-t001:** Auxiliary remote sensing data information.

Data	Temporal Resolution	Spatial Resolution
CYGNSS L1a	Daily	~0.5 km × 3.5 km (coherent)
CLDAS LST	Daily	0.625°
MCD43A3 reflectance	16 days	500 m
GPM IMERG Late Precipitation L3 precipitation	Daily	0.1°
ASTER surface elevation	/	90 m
SMAP surface roughness	Daily	9 km
SMAP VWC	Daily	9 km

**Table 2 sensors-23-09066-t002:** Feature importance values of different auxiliary variables.

Variables	Feature Importance
CYGNSS SRn	0.385
VWC	0.162
Slope	0.063
Surface roughness	0.026
Precipitation	0.130
Surface albedo	0.071
LST	0.143
Elevation	0.011
Aspect	0.009

## Data Availability

Data used in this study are available on request from the first author.

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
