# Peer review of "Daily Soil Moisture Retrieval by Fusing CYGNSS and Multi-Source Auxiliary Data Using Machine Learning Methods"

_sensors, 2023, doi:10.3390/s23229066_

Round 1

Reviewer 1 Report

Comments and Suggestions for Authors

It is of great value to study the method of daily soil moisture retrieval with CYGNSS and multi-source data. I have several comments and/or questions concerning the contents and the presentation of the manuscript which may help improve the publication.

1. Could the authors please provide more information about the type of GNSS receiver used in the experiment? Can the ordinary geodetic GNSS receivers be used? In Eq(1), gains of GNSS antenna and RHCP power of GNSS satellite are needed. How do the authors obtain these information?

2. It seems that only GPS signals are used. What the other GNSS systems, such as Galileo, GLONASS and BeiDou? Can they also be used for the retrieval of SM?

3. Accompanied with new data source, new errors will also be introduced. Do the authors need some additional data processing methods to alleviate or avoid the errors? If the purpose of data fusion is to improve the accuracy of SM retrieval, the authors may also need prove that the contribution of CYGNSS data is not negligible. Maybe, similar accuracy could be obtained only with multi-source auxiliary data. More comparisons or at least more discussion are needed to make the conclusions more persuasive.

4. In Section 3.1, the permutation_importance and feature importance are mentioned, and they are said to be calculated from the sklearn module of Python. I am wondering whether the authors can provide more detailed principles of the calculation. What is the relationship between feature importance and the commonly used correlation coefficient?

5. What does ubRMSE mean and how is it calculated? What the difference between ubRMSE and the commonly used RMSE?

6. Unlike Figure 4(a) and (b), the color bar in Figure 4(c) is discrete and the widths of bins are different from color to color, i.e. 0.11(=0.19-0.08) for red and 0.03 (0.08-0.05) for yellow. Could the authors please give some explanation for such design? By the way, the texts in both Figure 4 and 5 are not legible enough, and the resolutions of the figures can also be improved.

Comments on the Quality of English Language

7. The language are generally good, but also should be checked and improved all through the paper, to avoid typos such as “using by” in Line 178.

Author Response

We thank Reviewer #1 for her/his valuable time in reviewing this manuscript and providing insightful comments. We have carefully revised the manuscript to address the issues and comments raised by the Editor and reviewers. Point-to-point responses are listed in the attachment.

Reviewer 2 Report

Comments and Suggestions for Authors

See the attached, marked up version of the draft for line-by-line comments.

The authors present a soil moisture retrieval process for GNSS-R observables that uses multiple gradient boosted regression tree models to estimate soil moisture over difference land cover types. This analysis focuses on South Eastern China, comparing results (from a window of time temporally separated from training) to SMAP (the target variable) and an array of in situ soil moisture probes. The approach is reasonable, though not particular ground-breaking (multiple models have been used in similar retrievals before), and the 

Major issues are detailed below:

- Repeatedly the authors claim that only multiple "subnetworks" are capable of us making good retrievals due to landcover information. This is patently incorrect. You can easily build a single model that includes land cover information. There are several blanket statements like this that need to be revised as they haven't shown anything this definitive.

- There is a potential issue with their comparison to in situ data. They claim to use CYGNSS v3.0 data, which only extends back to August 1, 2018. They then make comparison to in situ data from July 2018. How can this be? Did they switch to using v2.1 data? If so, there are issues with that. They need to clarify this.

- The authors include very strong ancillary information (surface temperature and precipitation) that can act as the dominant source of information on model performance. They include a table that shows the relative importance of their auxiliary inputs, but do not include GNSS-R derived input in this comparison. They need to add that to the table to convince the reader that the model is learning from GNSS-R, not precipitation (or even surface temperature). 

- As an additional comment to the above, it is surprising that precipitation shows such weak importance in that table. In non-agricultural regions, all trends in SM must come from precipitation, so I would expect that input to be significant (it has been in all of our group's studies). The authors should be more specific with with GPM dataset they are using.

- There needs to be more detail related to the in situ data. Is this data publicly available? What was the match up method (nearest to within what distance? time?), how many samples went into calculating the presented numbers?

- The study described in section 4.1 is woefully insufficient to support the argument they are making (that they have chosen the best possible ML-model). First, the other models they are comparing to are barely described. Second, it's a wildly general statement that I don't think could be validated with even an entire paper dedicated to the question. My suggestion is to back off from the "this is the best possible way" (it is not) and show simply that it works.

Comments on the Quality of English Language

There are several instances of grammatical/structural issues highlighted/commented in the attached copy of the draft. A few key things to point out that could be English (or not), but should be changed:

- The word "optimal" is used in many locations where it is unnecessary or just wrong. Suggest reviewing and removing these cases.

- The multiple models in this paper are referred to as "subnetworks" throughout. This is not standard terminology for this type of model (these are decision trees, not neural networks). Suggest revising the discussion to refer to "multiple models" instead of subnetworks.

- Their definition of "semi-empirical" model is incorrect. A better description would be "data-driven" or "ML-based". 

Author Response

We thank Reviewer #2 for her/his valuable time in reviewing this manuscript and providing insightful comments. We have carefully revised the manuscript to address the issues and comments raised by the Editor and reviewers. Point-to-point responses are listed in the attachment.

Round 2

Reviewer 2 Report

Comments and Suggestions for Authors

The authors have addressed most of the concerns noted in my initial review. Attached is a marked up copy of the draft (with inline comments), but here are my two remaining issues to address:

The authors should explain why they chose to use v3.0 data (which has limited temporal range), instead of v2.1 or v3.1. Blending retrievals with these different version comes at a risk (the DDMs and derived products are inherently different), as they have done for comparison to in situ locations.

The f(theta) function needs additional description. A full derivation is not necessary, but the reader needs to understand what it is and why it is needed.

Comments on the Quality of English Language

I included a couple notes on where English issues exist.
